



# Construction of homogenized daily surface air temperature for Tianjin city during 1887-2019

Peng Si[1], Qingxiang Li[2, 4], and Phil Jones[3]

[1] Tianjin Meteorological Information Center, Tianjin Meteorological Bureau, Tianjin, China

[2] School of Atmospheric Sciences, Sun Yat-sen University, Guangzhou, China

[3] Climatic Research Unit, School of Environmental Sciences, University of East Anglia, Norwich, UK

[4] Key Laboratory of Tropical Atmosphere-Ocean System (Sun Yat-sen University), Ministry of Education, and Southern Laboratory of Ocean Science and Engineering (Guangdong Zhuhai), Zhuhai, China

**\*Corresponding author:**

Prof. Qingxiang Li

School of Atmospheric Sciences

Sun Yat-Sen University

Tangjiawan, Zhuhai Campus of SYSU

Zhuhai, China, 519082

Tel/Fax: 86-756-3668352

E-Mail: liqingx5@mail.sysu.edu.cn



**Abstract.** The century-long continuous daily observations from some stations are important for the
study of long-term trends and extreme climate events in the past. In this paper, three daily data sources:
(1) Department of Industry Agency of British Concession in Tianjin covering Sep 1 1890-Dec 31 1931
(2) Water Conservancy Commission of North China covering Jan 1 1932-Dec 31 1950 and (3) monthly
journal sheets for Tianjin surface meteorological observation records covering Jan 1 1951-Dec 31 2019
have been collected from the Tianjin Meteorological Archive. The completed daily maximum and
minimum temperature series for Tianjin from Jan 1 1887 (Sep 1 1890 for minimum) to Dec 31 2019 has
been constructed and assessed for quality control and an early extension from 1890 to 1887. Several
significant breakpoints are detected by the Penalized Maximal T-test (PMT) for the daily maximum and
minimum time series using multiple reference series around Tianjin from monthly Berkeley Earth,
CRUTS4.03 and GHCNV3 data. Using neighboring daily series the record has been homogenized with
Quantile Matching (QM) adjustments. Based on the homogenized dataset, the warming trend in annual
mean temperature in Tianjin averaged from the newly constructed daily maximum and minimum
temperature is evaluated as $0.154 \pm 0.013$ ℃ decade$^{-1}$ during the last 130 years. Trends of temperature
extremes in Tianjin are all significant at the 5% level, and have much more coincident change than those
from the raw, with amplitudes of -1.454 d decade$^{-1}$, 1.196 d decade$^{-1}$, -0.140 d decade$^{-1}$ and 0.975 d
decade$^{-1}$ for cold nights (TN10p), warm nights (TN90p), cold days (TX10p) and warm days (TX90p) at
the annual scale. The adjusted daily maximum, minimum and mean surface air temperature dataset for
Tianjin city presented here is publicly available at https://doi.pangaea.de/10.1594/PANGAEA.924561
(Si and Li, 2020).

## 1 Introduction

Instrumental observation records at meteorological stations are the most widely used first-hand information about weather and climate change and variability. They have the advantages of representativeness as well as accuracy compared to other data (Leeper et al., 2015; Dienst et al., 2017; Xu et al., 2018). The most representative long-term observational temperature datasets in the world since IPCC AR5 (2013) includes: Global Historical Climatology Network(GHCN)-monthly(GHCNm) dataset (Lawrimore et al., 2011; Menne et al., 2018), Climatic Research Unit (CRU) datasets (Jones et al., 2012; Harris et al., 2020), Goddard Institute for Space Studies (GISTEMP) dataset (Hansen et al., 2010; Lenssen et al., 2019) and Berkeley Earth Surface Temperature (BEST) (Rohde et al, 2013). Recently, in order to make up for the limited coverage and the potential regional variability of data quality of current global climate datasets, Xu et al. (2018) has developed a new dataset of integrated and homogenized global land surface air temperature-monthly (C-LSAT). This has been updated to C-LSAT2.0, with the data extended to the period 1850-2019 (Li et al., 2020a; 2020b). These datasets were all developed at the monthly scale based upon meteorological station records from different continents over the world through the integration of different data sources, quality control of climate outliers, time and space consistency, and the analysis of data homogenization. The Global Historical Climatology Network-daily (GHCNd) dataset has also been developed, to meet the needs of climate analysis and monitoring research, but about two-thirds of the stations contributing to this dataset report precipitation only. In addition, GHCNd dataset has not been homogenized for artifacts due to changes in reporting practice at different times at particular stations (i.e., systematic biases), although the entire dataset has been quality controlled (Menne et al., 2012).

Chinese scholars, since the 1980s, have also carried out many studies on the establishment of long-term observational time series in China, but they often mainly used tree rings, ice cores, historical materials and other proxy data as part of the restoration of time series before the 1950 (the founding of the People's Republic of China) (Wang et al., 1998; 2000; Zheng et al., 2015; Yu et al., 2018). The



results based on these data are of great significance as they reveal the characteristics of climate
periodicity and multi-scale changes over the past hundred years, but they are insufficient to meet the
needs of quantitative monitoring and detection of long-term extreme climate events. In particular, there
are many limitations when homogenizing the time series before the 1950s (including the establishment
of reference series) such as the lack of continuous observational data, detailed and reliable metadata
information, leading to the increase of uncertainties for regional and/or local climate analysis (Li et al.,
2020c). As a result there still exists many uncertainties in the characteristics of climate change from the
19[th] century to the mid-20th century (Li et al., 2010; 2017; Sun et al., 2017; Cao et al., 2017).
Since daily time series generally contain many more observations than monthly or annual series,
daily analyses potentially have greater precision. As a result they are more useful in climate trend and
variability studies, especially for extreme events (Vincent et al., 2012; Xu et al., 2013; Trewin, 2013;
Hewaarachchi et al., 2017). However, due to difficulties in collecting and/or receiving daily data all over
the world as well as non-climatic effects such as changes in observation times, there are numerous issues.
For example, observations from temperature sites at principal stations in Canada were changed to be
read at 0000UTC to 0600UTC (Vincent et al., 2002), making it is very difficult to form a global daily
data product at century-long scales. This makes it extremely difficult to study global and/or regional
extreme events over the past hundred years, especially before 1950. For some regional areas, daily
instrumental observations may be extended to the 19[th] century and hence they are more valuable. Png et
al. (2020) has compiled 463,530 instrumental observations of daily temperature, precipitation and
sunshine from 319 stations distributed over China during 1912-1951 mostly from the source of monthly
reports from the Institute of Meteorology, Nanjing; observatories over Japan-occupied Manchuria and
the Japanese Army for North China. Since this is a daily data, it is immensely useful for the analysis of
mesoscale and sub-seasonal climate variations. Although the earliest instrumental observations in China
began in the 1840s, observations at some sites were interrupted during 1940s due to wars (e.g. the War
of Resistance Against Japan and the War of Liberation) and hence many pieces of information have





likely been lost. Studies of the rescuing, processing and constructing complete and continuous daily site
data over China are somewhat rare.

Due to the historic reasons of leased territory in China, some local single sites often have multiple

observational sources before 1950. For example, for Qingdao, monthly surface air temperature series
during 1899 - 2014 have been constructed based on newly digitized and homogenized observations from
the German National Meteorological Service from 1899 to 1913 (Li et al., 2018). Tianjin meteorological
station is one of the typical stations with more than one hundred years of observed climate data in China
(Yan et al., 2001; Si et al., 2017). This station also has multi-source observations before 1950 as
observed at some other meteorological stations in China having century-long datasets. Thus, considering
Tianjin station as an example, this paper aims to construct a new daily instrumental maximum and
minimum temperature series on the century scale in China, through integration, quality control,
extension and homogenization of the multiple daily observations. The newly constructed daily
temperatures in Tianjin provide relatively longer, more complete and reliable climate series for studies of
climate and extreme climate change over century-long scales.

The remainder of this paper is arranged as follows: Section 2 describes the station histories from

Tianjin. The basic data and reference data sources and their pre-processing are introduced in section 3.
Section 4 introduces the procedures of constructing new daily maximum and minimum temperature
series. Section 5 presents average and extreme temperature trend change based on newly constructed
series. The availability of the resulting dataset (Si and Li, 2020) is reported in section 6, and summary of
results and some discussion are given in section 7.
**2    Historical evolution of Tianjin meteorological observation station**
Wu (2007) showed that Tianjin meteorological observation station was under the control of the
Department of Industry Agency of British Concession in Tianjin covering September 1887 to December
1941. During the period from September 1904 to December 1949, it was co-ordinated by many


departments, such as Japan Central Meteorological Station, Central Meteorological Bureau of the
Republic of China, Aviation Department of North China Military Region of the People's Liberation
Army (PLA), Shunzhi Water Conservancy Commission, Water Conservancy Commission of North
China and Water Conservancy Engineering Bureau of North China. However, only the records of daily
maximum and minimum temperatures from Department of Industry Agency of British Concession in
Tianjin (Fig.1a-b) and Water Conservancy Commission of North China (Fig.1c) have been collected by
the Tianjin Meteorological Archive. Each of these is continuous and complete, and most importantly
they can be connected to each other on 1 Jan 1932 without overlapping, thereby forming a complete and
continuous time series before 1950.

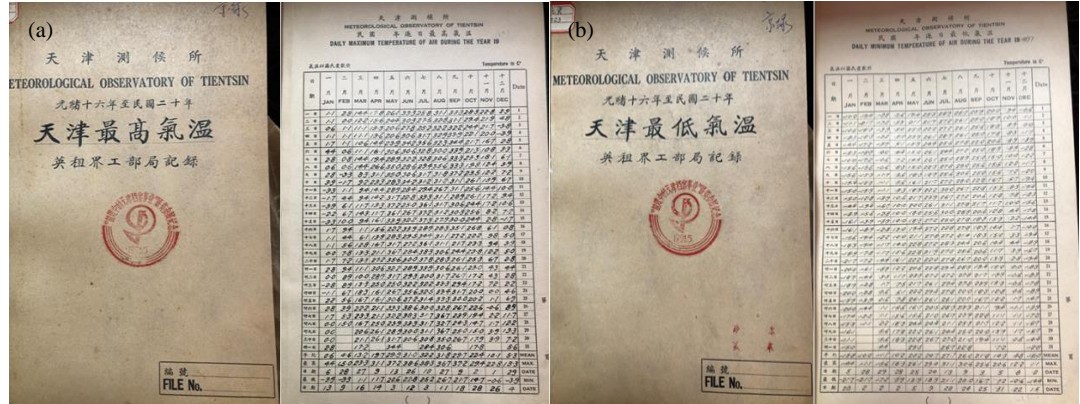


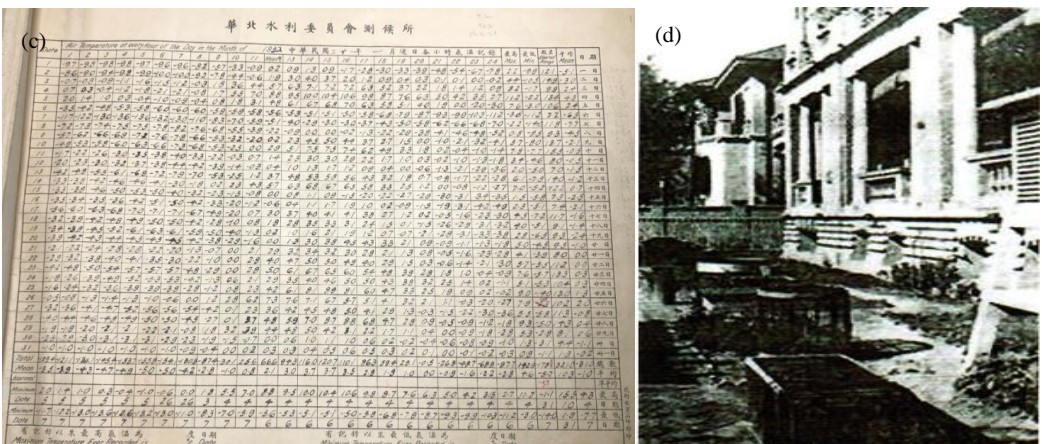



**Figure 1.** The handwritten observation records of Tianjin before 1950, (a) and (b) for records of maximum and
minimum temperatures from Department of Industry Agency of British Concession in Tianjin, (c) for records from
Water Conservancy Commission of North China, and Tianjin Observatory at No. 22 Ziyou Road (d). These
photographs were provided by Tianjin Meteorological Archive, Tianjin Meteorological Bureau.
The daily temperature records at Tianjin meteorological observation station that we have accessed
begin with Sep 1 1890 collected by Tianjin Meteorological Archive (Fig.1a-c). The metadata history of
the Tianjin observation station during Sep 1 1890 - Dec 31 2019 is listed in Table 1. The history is sorted
according to the Surface Meteorological Observation Specifications in China (versions of 1950, 1954,
1964, 1979, and 2003), metadata of China surface meteorological station and journal sheets of Tianjin
surface meteorological records. Changes to observational times have been marked on the original time
series of maximum and minimum temperatures for Tianjin city (see Fig. 2). As shown in Table 1, Tianjin
observation station has relocated four times since 1890, several times in 1921 (Fig. 1d), Jan 1 1955, Jan
1 1992, and Jan 1 2010 without prominent changes in elevations. The environment surroundings of
Tianjin station changed from urban to suburban in 1955, accompanied by a number of instrument
changes. In this period, changes to the instrument manufacturer have happened four times for both
maximum and minimum temperature series, as well as changes of automatic observation instead of
manual observation in 2004 and a new generation replacement of last automatic instrument in 2014. In
documented metadata (Table 1), there have been changes of observing time four times for both
maximum and minimum temperatures since 1951, but they were always recorded over a 24-hour
observational window at Beijing Time (BT) or similar to BT (as for the period of Jan 1 1954 - Dec 31
1960). Moreover, it is important to mention here that since 1951 there were two stations viz., Tianjin and
Xiqing collocated in the Tianjin area. Due to rapid urbanization at the surrounding environmental Tianjin
area, the old Tianjin site gradually becomes less representative as a climate observation station and
therefore since Jan 1 1992 afterwards, observations at Xiqing station are used to replace the old Tianjin

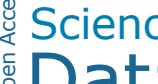

station. This can also be considered as Tianjin station being relocated to Xiqing station since then.
**Table 1.** The history logs at Tianjin meteorological observation station during Sep 1 1890 - Dec 31 2019.

| Periods | Latitude | Longitude | Altitude (m) | Address (Surrounding environment) | Relocation description | Instrument change | Observing time |
|---|---|---|---|---|---|---|---|
| 1890.9.1— 1921 | 39°07' | 117°12' | unspecified | unspecified | —— | unspecified | unspecified |
| 1921— 1950.12.31 | 39°08' | 117°11' | 6.0 | No. 22 Ziyou Road, The Third Distribution, Tianjin (urban) | unspecified | unspecified | unspecified |
| 1951.1.1— 1953.12.31 | 39°08' | 117°11' | 6.0 | Same as above | —— | —— | Tmax 18:00 Tmin 09:00 |
| 1954.1.1— 1954.12.31 | 39°08' | 117°11' | 6.0 | Same as above | —— | Tmax 1954.1.1 Tmin 1954.1.1 | unspecified |
| 1955.1.1— 1960.12.31 | 39°06' | 117°10' | 3.3 | Zunyi Road, Hexi Distribution, Tianjin (suburban) | 5 km north of the original site | —— | unspecified |
| 1961.1.1— 1991.12.31 | 39°06' | 117°10' | 3.3 | Qixiangtai Road, Hexi Distribution, Tianjin (suburban) | —— | Tmax 1961.1.1; 1973.1.1; 1989.1.1 Tmin 1961.1.1; 1966.1.1; 1989.1.1 | 20:00 |



| 1992.1.1—2003.12.31 | 39°05' | 117°04' | 2.5 | Xidawa, Xiqing Distribution, Tianjin (suburban) | unspecified | —— | 20:00 |
|---|---|---|---|---|---|---|---|
| 2004.1.1—2009.12.31 | 39°05' | 117°04' | 2.5 | Same as above | —— | automatic observation | pick up from timing minutes data |
| 2010.1.1—2013.12.31 | 39°05' | 117°03' | 3.5 | Jingfu Road, Xiqing Distribution, Tianjin (suburban) | 1.5km southwest of the site in 1992 | automatic observation | pick up from timing minutes data |
| 2014.1.1 to now | 39°05' | 117°03' | 3.5 | Same as above | —— | new generation of automatic observation equipment | pick up from timing minutes data |

The straight line (——) indicates no change. The observing time is at Beijing Time (BT). Tmax and Tmin indicate the
maximum and minimum temperature, respectively.

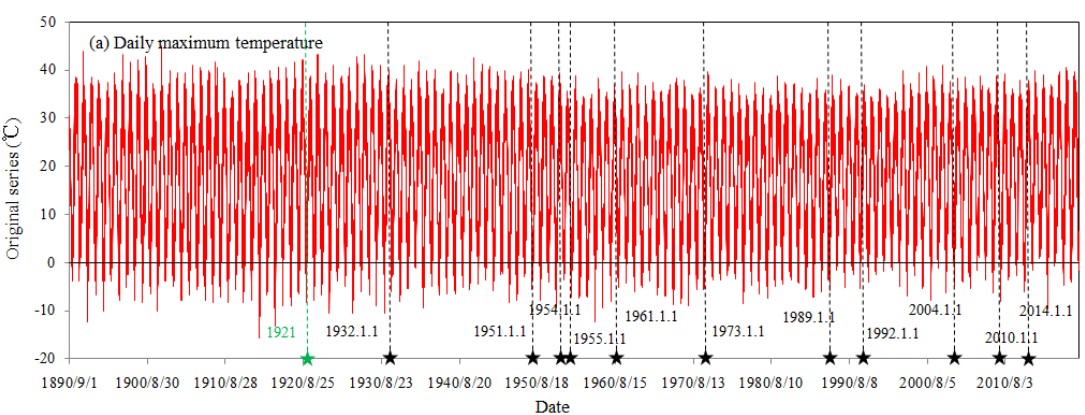



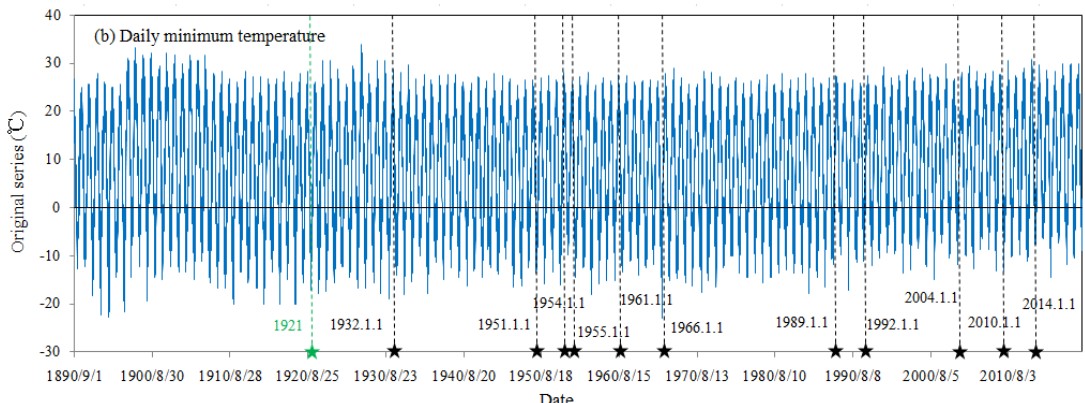

**Figure 2.** Original series of daily maximum temperature (a) and minimum temperature (b) at Tianjin meteorological
observation station covering Sep 1 1890 - Dec 31 2019. Black stars with vertical dashed lines on the axes mark
metadata times (The relocation point in 1921 is marked by green star with vertical dashed line due to no specific date).

## 3 Data sources

### 3.1 Original data and preliminary quality control

Based on the analyses of metadata history at Tianjin station discussed in section 2, this paper selects
three observation temperature data sources collected by Tianjin Meteorological Archive as the basic data
to construct the daily maximum and minimum temperature time series at Tianjin station from 1890 (Fig.
2). These are the daily observation records from (1) Department of Industry Agency of British
Concession in Tianjin covering Sep 1 1890 - Dec 31 1931; (2) Water Conservancy Commission of North
China during Jan 1 1932 - Dec 31 1950; and (3) monthly journal sheets of Tianjin surface
meteorological observation records covering Jan 1 1951 - Dec 31 2019. Since there are no missing data
and overlap for each of the three daily sources, the three daily data resources are directly spliced into a
complete time series. However, in view of the regime changes of operation and different station numbers
between Tianjin station and Xiqing station that happened at Jan 1 1992 (Table 1), the daily records from
Jan 1 1951 to Dec 31 1991 observed in the Hexi area and those observed at Xiqing area from Jan 1 1992
to Dec 31 2019 are used to form the basic daily time series from Jan 1 1951 to Dec 31 2019 for the long





series.

A preliminary quality control procedure consisting of multiple steps was carried out on the original

integrated daily time series of maximum and minimum temperature from Sep 1 1890 to Dec 31 2019 at
Tianjin to remove any errors caused by manual observations, instrument malfunctions and digital inputs.
Firstly, the range of the daily maximum or minimum temperature was scrutinized for magnitude beyond
the limit of 60 ℃ and -80 ℃ as errors. Fortunately, both the series have no such error. Secondly based
on anomalies from the 1961-1990 as reference period, climatic outliers of maximum and minimum
temperature are assessed considering a magnitude exceeding five standard deviations of their monthly
anomalies as outliers during 1890-2019. There is no outlier found in our validation. Finally, internal
consistency is investigated by checking if there is any minimum temperature data greater than or equal
to the maximum at the same date and no such inconsistencies were found. It is important to mention that
there is a sudden rise in annual minimum temperature series during the year 1927 even after these three
checks. The offsets of the discontinuities in 1927 compared with averages for the two sections before
and after it are 4.2 ℃ and 3.4 ℃, respectively. We repeated the steps of outlier checks once by
reviewing the earlier condition with three times the standard deviation of monthly anomalies. Results
indicate that most of the daily minimum data for April to October 1927 exceed the current condition and
finally the data during this period so were set to missing values. Even though, the quality of original
daily maximum and minimum temperatures during 1890 - 2019 at Tianjin station is good, these checks
provide a good foundation for the subsequent construction of a reliable homogenized daily series.
**3.2   Reference data**
Wu (2007) documented that although the earliest surface observation records at Tianjin station start with
the year 1887, the observed daily maximum and minimum temperatures collected by Tianjin



Meteorological Archive began with September 1890 (Fig. 2). Therefore, some additional reference data
sources are selected to extend the daily temperature series from January 1887 to August 1890 and
lengthen the established daily temperature data to as early as possible. In addition, it is extremely
important to establish an objective as well as a reasonable reference series for data homogenization. But
due to non-availability of observation records and station metadata before 1950 especially for daily data,
it is impossible to find a complete and reliable observed temperature series as a reliable reference series
for Tianjin. Based on few recently reported studies (Li et al., 2020a; Lenssen et al., 2019; Xu et al., 2018;
and Menne et al., 2018), we employ the station series or the interpolated temperature series using
neighboring grid boxes from three global land surface temperature observation series (Table 2) as
reference data sources for extension and establishment of reference data series used in data
homogenization at Tianjin station. Plots of the 'Tianjin' station from all three series are shown in Figure
3. The three global Land Surface Temperature (LSAT) are (1) Berkeley Earth land temperature
(Berkeley Earth; Rohde et al., 2013; http://berkeleyearth.org/data/); (2) Climatic Research Unit (CRU)
Time-Series     (TS)     version     4.03     (CRUTS4.03;     Harris     et     al.,     2020;
http://data.ceda.ac.uk/badc/cru/data/cru_ts/cru_ts_4.03/data/) and (3) Global Historical Climatology
Network     (GHCN)     version3     (GHCNV3;     Lawrimore     et     al.,     2011;
https://www.ncdc.noaa.gov/ghcnd-data-access).
The selected three LSAT are not independent as they likely use common input observations. The
multiple datasets provides a variety of useful checks because they employ different ways of handling
data problems such as incomplete spatial and temporal coverage and non-climatic influences on
meteorological measurement (Hansen et al., 2010). As shown in Table 2, the three LSAT involve quality
control and homogenization but using different methods. The records of Berkeley Earth were usually





split into portions occurring before and after known and presumed discontinuities (e.g., from station
relocation or instrument changes) without adjustment. For CRUTS4.03, most of these data have been
adjusted, because the ultimate sources of most station records are from National Meteorological Services
(NMSs), so China Meteorological Administration (CMA) for Tianjin. GHCNV3 station data used in this
paper are the quality controlled and adjusted (QCA) data, which were produced by the developers of
GHCN-Monthly. Two types of grid data, CRUTS4.03 and Berkeley Earth are both interpolated to the
site level using the bilinear method.

From Table 2, only Berkeley Earth daily maximum/minimum temperatures are available. So the

maximum temperature Berkeley Earth-daily data corresponding to the site level Tianjin station is
selected as the extension data for the period of Jan 1 1887-Aug 31 1890 (The specific information about
extending the processing for this period is in supplemental material), and the daily minimum series still
begins with the date on Sep 1 1890 due to scarcity of reference data sources.

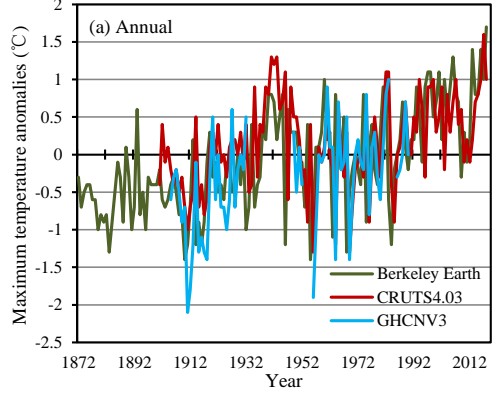 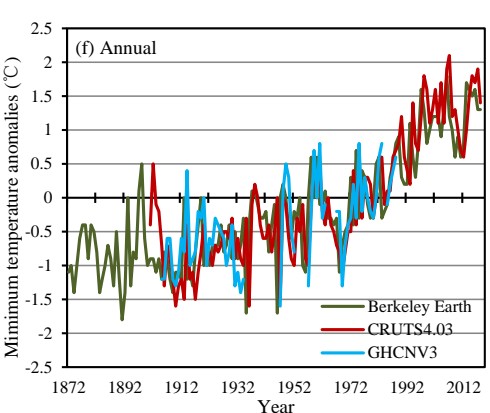









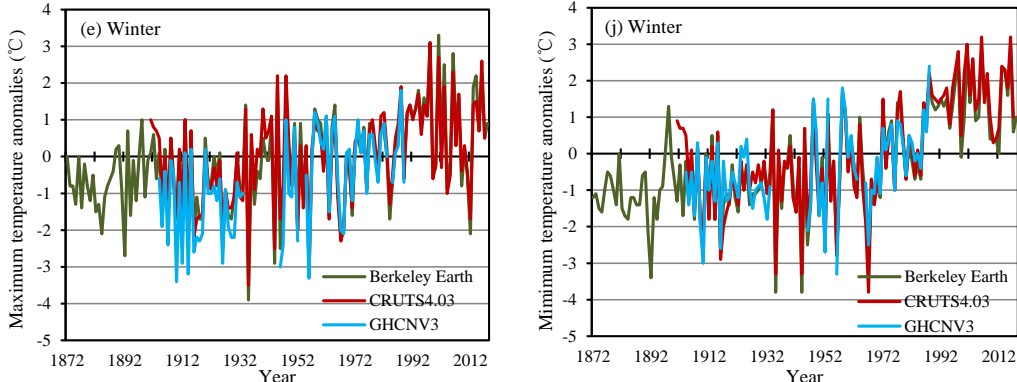


**Figure 3.** The annual and seasonal average anomalies of maximum (a-e) and minimum (f-j) temperatures based on the interpolated series of Berkeley Earth and CRUTS4.03 and the station series of GHCNV3 for Tianjin station from 1961-1990 base period.

**Table 2.** Information of reference data sources.

| Data sources | Monthly series | Daily series | Gridded data | Station data | Temporal resolution used here | Gridded or station data used here | Time periods only for Tianjin in situ level | Units | Quality control | Adjustment |
|---|---|---|---|---|---|---|---|---|---|---|
| CRUTS4.03 | √ | × | √ | √ | monthly | 0.5 °×0.5 °gridded | 1901.1-2018.12 | ℃ | √ | √ |
| Berkeley | √ | √ | √ | √ | monthly | 1 °×1 °gridded | 1872.12-2019.12 | ℃ | | |
| Earth | √ | √ | √ | × | daily | 1 °×1 °gridded | Tmax1880.1-2018.12 | ℃ | √ | × |
| | | | | | | | / Tmin1903.1-2018.12 | | | |
| GHCNV3 | √ | √ | √ | √ | monthly | station data | 1904.1-1990.12 | ℃ | √ | √ |

## 4 Construction of daily maximum and minimum temperature series from 1887 to 2019

On the basis of the quality controlled and the extended series, the daily homogenized maximum and
minimum observation temperature series in Tianjin were constructed by means of the flow chart
illustrated in the dashed-box part of Fig. 4. Homogenization is an important step to eliminate the
discontinuities in observation records induced by non-climatic influences such as station relocation,
instrument change, observing time change and so on. Most importantly, the true characteristics of
climate change are preserved in this process (Quayle et al., 1991; Della-Marta and Wanner, 2006;
Haimberger et al., 2012; Rahimzadeh and Zavareh, 2014; Hewaarachchi et al., 2017).

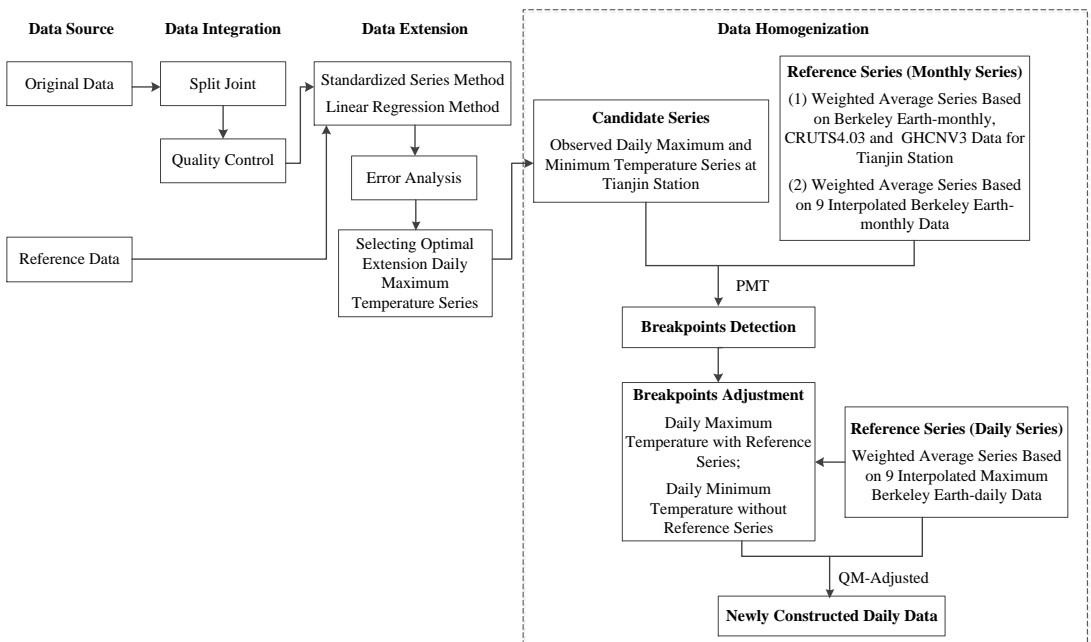


**Figure 4.** The flow chart of construction for century-long daily maximum and minimum temperature series in Tianjin.
## 4.1 Establishment of the reference series
In the process of homogenization, reasonable reference series plays an important role in the reliability of
the detected breakpoints. So in this section, we will establish monthly and daily reference series for the
maximum or minimum temperature series at Tianjin station and use them for breakpoint detection and
adjustment, respectively. Both reference series are established using a weighted average method. For
monthly reference series, we will establish two types in order to make the detected breakpoints more
reasonable and reliable. First, reference data are based on the combination of the interpolated
temperature series from Berkeley Earth-monthly and CRUTS4.03 and station series from GHCNV3 data
for Tianjin site (the three global LSAT datasets) and secondly based on the interpolated temperature
series from Berkeley Earth-monthly data only. From the three LSAT data, the weight coefficients are the
square of the correlation coefficients between each LSAT and Tianjin's observed data. The daily



reference series we use is based on the interpolated temperature series from Berkeley Earth-daily data
only.

In the case of the interpolated temperature series from Berkeley Earth-monthly or daily data only,

the site level data are derived from the station network across the Beijing-Tianjin-Hebei area in China
(Fig. 5). These stations are selected as follows: Firstly, the potential stations less than 300km at
horizontal distances around Tianjin station and with altitude differences within 200m are chosen;
Secondly, we will select 10 stations those are closest to Tianjin station using a spherical distance; Finally,
9 stations are confirmed which are consistent between step 1 and step 2. In Figure 5 (the right), these 9
are identified by green solid circles with black or red stars. Thus, the interpolated temperature series
from Berkeley Earth-monthly or daily reference series are generated using the weighted average of the 9
stations. These weights are calculated as the square of correlation coefficients between the interpolated
temperature series from Berkeley Earth-monthly or daily data for each 9 stations and Tianjin's observed
data. Recall also that, the missing values in the original daily minimum temperature at Tianjin for April
to October 1927 (checked in section 3.1) were replaced by the corresponding data from the weighted 9
interpolated temperature series from Berkeley-daily data.

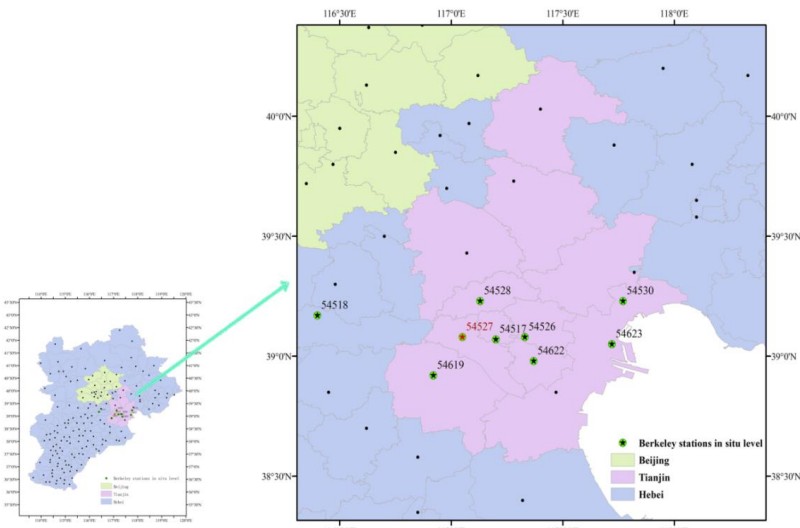

**Figure 5.** Geographical distribution of the surface observation stations (black solid circles) at Beijing-Tianjin-Hebei area in China and the selected 9 stations (green solid circles with black or red stars).

## 4.2 Breakpoints detection and adjustment

The RHtestsV4 software package is used to homogenize the daily maximum and minimum temperature data at Tianjin station. The software consists of the Penalized Maximal T-test (PMT) (Wang et al., 2007) and Quantile Matching (QM) adjustment (Wang et al., 2010), both of which are used to detect and adjust the known or presumed discontinuities. As observed in earlier reported studies (Vincent, 2012; Trewin, 2013; Xu et al., 2013), homogenization at the daily timescales is much more challenging than that at monthly or annual scales. Thus, firstly we test Tianjin's monthly observed maximum and minimum temperature series averaged from the daily ones to find the significant breakpoints by means of PMT at the 5% significance level using two types of monthly reference series. We then adjust the daily series at Tianjin station by QM-adjustment with or without the daily reference series.

The breakpoints in the segment before 1921 are mainly determined by objective judgment from the same shifts at the two monthly timescales simultaneously due to the scarcity of station metadata. Those



after 1921 are additionally assessed together with the station metadata and PMT detection at the 5%
significance level. According to Table 1, we made a list containing some possibilities that could cause
shifts in Tianjin's daily maximum and minimum temperature series (Fig. 2 vertical dashed lines). The
date of Jan 1 1932 and Jan 1 1951 are connection points for different data sources, 1921, Jan 1 1955, Jan
1 1992 and Jan 1 2001 indicate station relocations, and the others are the times of instrument and/or
observing time change. However, due to statistical non-significance, those potential discontinuities are
not considered as the final discontinuities (Fig. 6). So potential dates do not always cause artificial
discontinuities at the joining of the three observation segments for daily maximum or minimum
temperature series. Also all the instrument changes that happened for maximum and minimum series
have also not introduced any significant shifts. In this regard, they do not look like the changes that
happened with other networks around the world, such as the U.S. Cooperative Observer Program
(COOP) network (Leeper et al., 2015). Moreover, different observing times (including automated
observation system) also do not introduce any significant biases to the temperature time series, since the
daily maximum and minimum temperatures are always recorded over a 24-hour observational window.
Additionally, various versions of the surface meteorological observation specifications in China (e.g.
versions of 1950, 1954, 1964, 1979 and 2003) imply that the observation principles of the highest and
lowest thermometers are consistent, although there were a number of alterations of observing times.

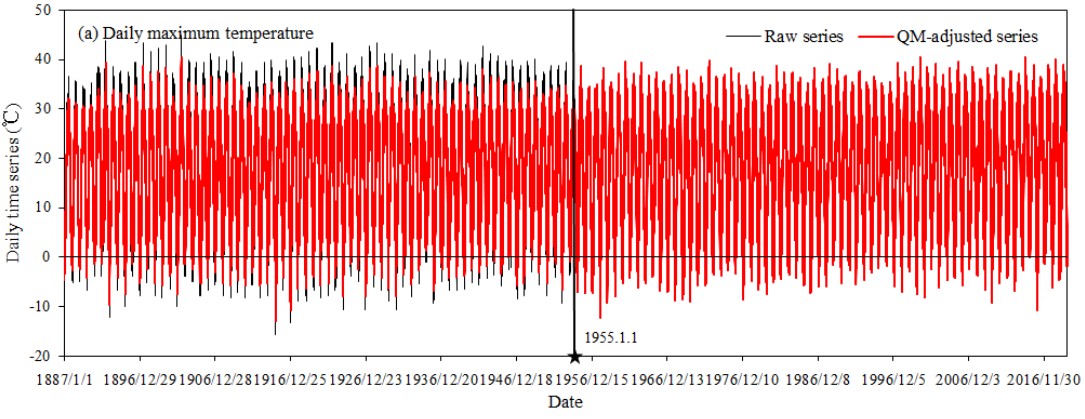


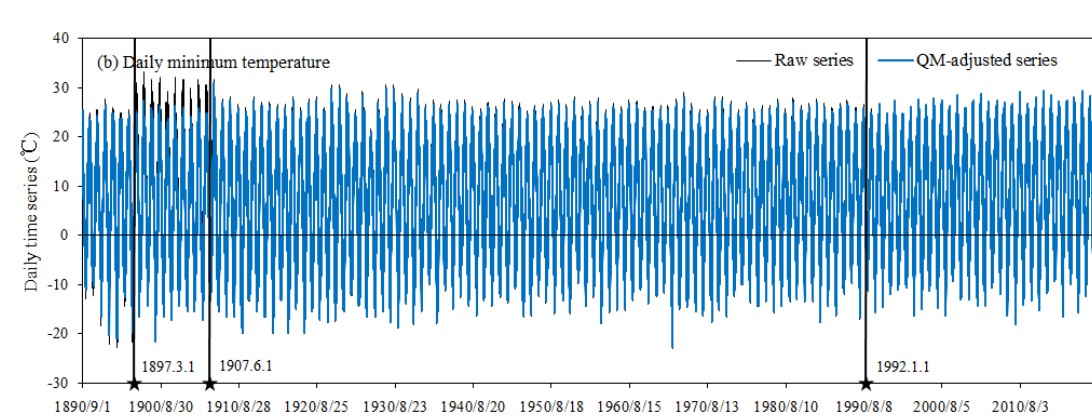


**Figure 6.** QM-adjusted and raw series (after quality control and extension) of daily maximum temperature (a) and
minimum temperature (b) at Tianjin meteorological observation station covering Jan 1 1887 (Sep 1 1890 for minimum)
to Dec 31 2019. Vertical solid lines demarcate the discontinuities at times Mar 1 1897, Jun 1 1907, Jan 1 1955 and Jan 1
1992.

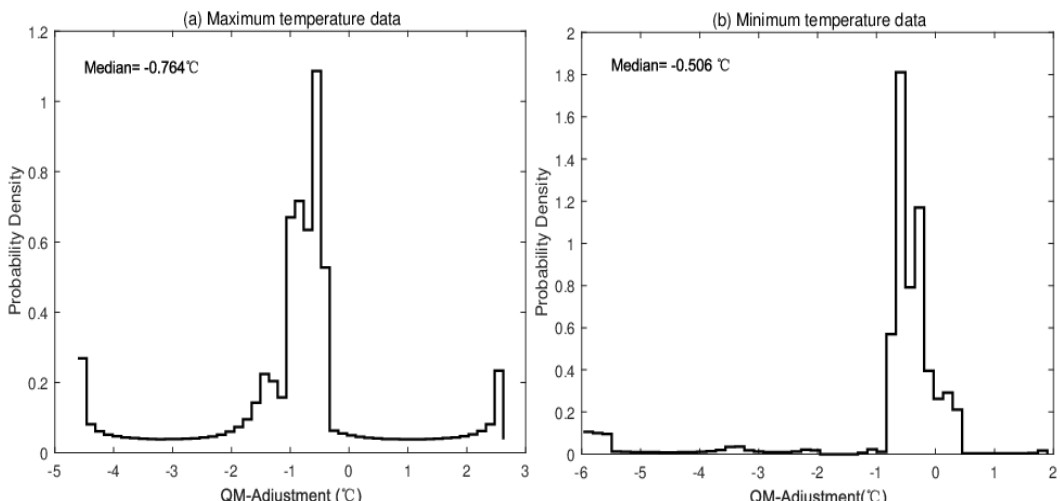

**Figure 7.** The amplitudes of QM adjustment applied to daily maximum (a) and minimum (b) temperature data at Tianjin meteorological observation station.

**Table 3.** The average QM adjustments at the monthly timescales applied to daily maximum and minimum temperature data at Tianjin meteorological observation station (Units: ℃).

|  | Jan | Feb | Mar | Apr | May | Jun | Jul | Aug | Sep | Oct | Nov | Dec |
|---|---|---|---|---|---|---|---|---|---|---|---|---|
| Maximum temperature | 1.136 | 0.246 | -0.616 | -0.687 | -1.322 | -2.484 | -2.817 | -2.285 | -1.046 | -0.590 | -0.582 | 0.566 |
| Minimum temperature | -0.105 | -0.297 | -0.634 | -0.979 | -1.084 | -1.090 | -1.207 | -1.184 | -1.050 | -0.951 | -0.624 | -0.317 |

Breakpoints in Mar 1 1897 and Jun 1 1907 are without metadata support, but those at Jan 1 1955 and Jan 1 1992 are confirmed by metadata of station relocation. The significant breakpoints are given in Fig. 6 as vertical solid lines. The amplitudes of QM adjustment applied to each individual daily maximum and minimum temperature data are [-4.606, 2.621 ℃] (Fig. 7a) and [-5.972, 1.897 ℃] (Fig. 7b). The medians of QM adjustment are -0.764 ℃ and -0.506 ℃ respectively. As shown in Fig. 7, there are about 75% of adjustments are covering -2.5～0.8 ℃ in daily maximum series. For the minimum ones, there are about 85% of adjustments are covering -0.8～0.5 ℃. Table 3 provides the average amplitudes of QM adjustment at the monthly timescales. It shows that for the maximum data, the larger positive

adjustments are mainly applied to series in January and December, while the larger negative adjustments
are mainly in June, July and August. For the minimum data, all the average amplitudes of QM
adjustment at the monthly timescales are negative, what is the same characteristic with the maximum
ones, temperature series in the warm months (e.g. June, July and August) are adjusted with larger
negative amplitudes, but smaller negative amplitudes for the series in the colder months (e.g. January,
February and December).

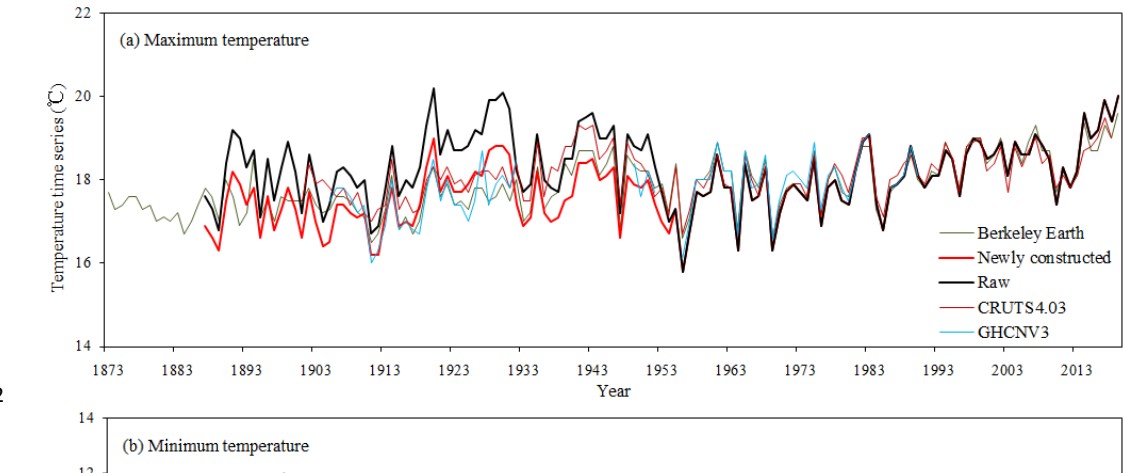


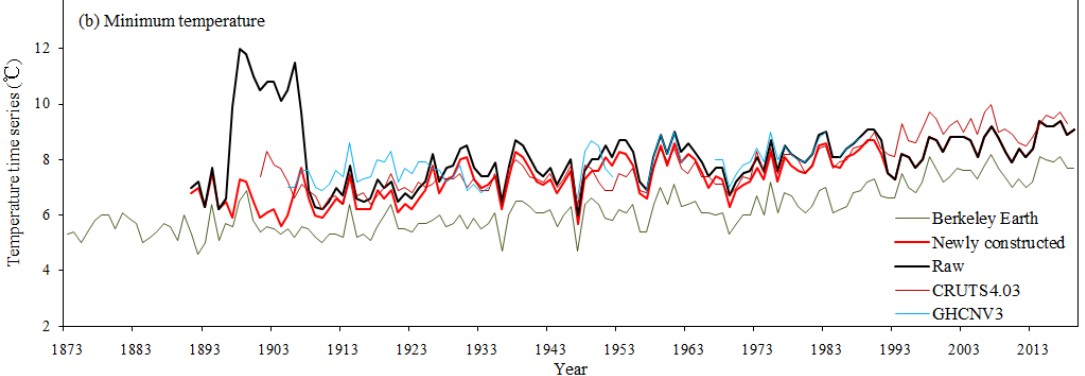


**Figure 8.** Annual average series of maximum (a) and minimum (b) temperatures in Tianjin from 1887 (1891 for
minimum) to 2019 based on newly constructed and raw daily data (after quality control and extension),
correspondingly with annual averaged data based on the interpolated series from Berkeley Earth-monthly (1873-2019)
and CRUTS4.03 (1901-2018) and station series from GHCNV3 (1905-1990) for Tianjin station.

The average annual maximum and minimum temperature series based on the adjusted daily data

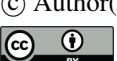



(the red lines) in Tianjin are given in Fig. 8. The raw time series for Tianjin (after quality control and
extension), the interpolated series from Berkeley Earth and CRUTS4.03 and station series from
GHCNV3 for Tianjin station averaged from their monthly data are also displayed in Fig. 8. This shows
that the newly constructed time series has removed the large shifts in maximum and minimum
temperature series before 1955 and 1992 (Fig. 8 red lines) compared with the raw ones (Fig. 8 black
lines). Especially for the minimum data, the QM-adjustments have offset the shifts between 1896 and
1908 to the greatest extent. Meanwhile, the newly constructed temperature data has similar inter-annual
variability and trend changes compared to those of Berkeley Earth, CRUTS4.03 and GHCNV3 during
the overlapping period.
**Table 4.** Definition of temperature extremes (Zhang et al., 2011). Days with maximum or minimum temperature above
the 90th and below the 10th percentiles are relative to the reference period of 1961 - 1990.

| Index | Description | Definition | Units |
|---|---|---|---|
| TN10p | Cold nights | Days when daily minimum temperature <10th percentile | days |
| TN90p | Warm nights | Days when daily minimum temperature >90th percentile | days |
| TX10p | Cold days | Days when daily maximum temperature <10th percentile | days |
| TX90p | Warm days | Days when daily maximum temperature >90th percentile | days |

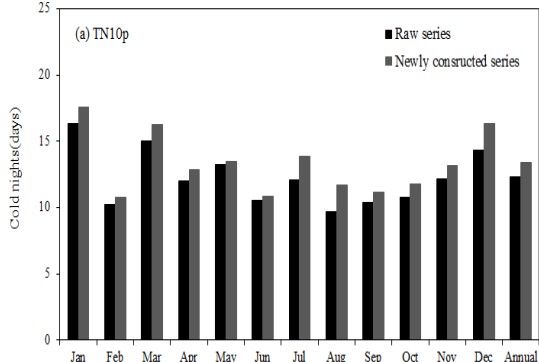

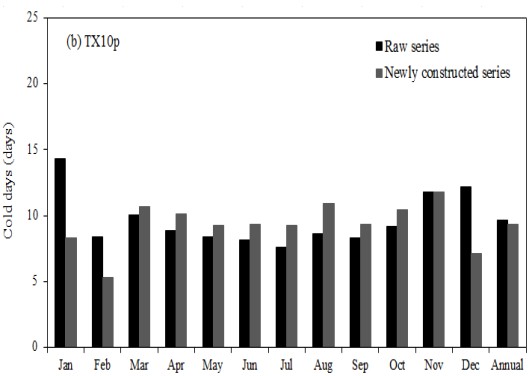


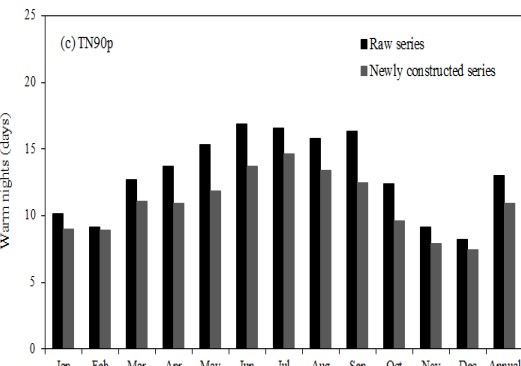
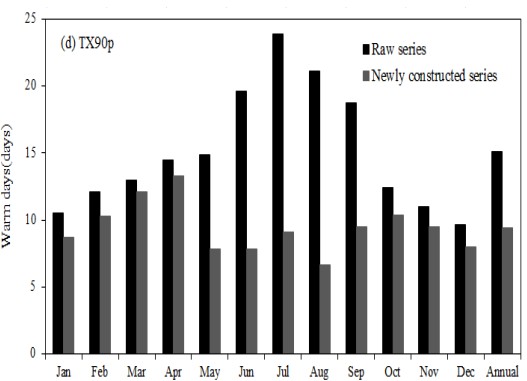


**Figure 9.** Annual and monthly temperature extremes of Cold nights (TN10p) (a), Cold days (TX10p) (b), Warm nights (TN90p) (c) and Warm days (TX90p) (d) for daily newly constructed and raw temperatures (after quality control and extension) in Tianjin.

Table 4 provides the definition of temperature extremes (Zhang et al., 2011). They are calculated based on the newly constructed and raw series (after quality control and extension) in Tianjin. As shown in Fig. 9, the number of TN10p (Fig. 9a) and TX10p (Fig. 9b) at the monthly timescales are increased by 0.3-2.0 days and 0.6-2.3 days based on the newly constructed series, especially in August they are increased by 2.0 and 2.3 days respectively. TX10p in cold months (January, February and December) from newly data are all less than those from the raw data, with the number of days decreased by 3.1-6.1. This is mainly due to large positive adjustments applied to daily maximum temperature in these months (Table 3). In the opposite sense, the number of TN90p (Fig. 9c) and TX90p (Fig. 9d) are decreased by 0.2-3.8 days and 0.9-14.8 days respectively based on the newly constructed series. The decreasing numbers of the two indices between May and September are prominent, especially for TX90p from June to August the number is decreased by 11.8-14.8 days. This is due to the large negative adjustments applied to daily maximum temperature in these months (Table 3). The number of TN10p (Fig. 9a) from newly constructed series at the annual timescales are increased by 1.1 days compared to the raw ones while for TX10p (Fig. 9b), TN90p (Fig. 9c) and TX90p (Fig. 9d) are decreased by 0.3, 2.1 and 5.7 days



respectively.

## 5    The temperature change trend in Tianjin based on newly constructed series

### 5.1    Mean temperature trend during the last 130 years

**Table 5.** Comparisons between newly constructed surface air temperatures and previous assessments of the annual trend change at the century-long scale in Tianjin with uncertainties at 95% significance level (Units: ℃ decade$^{-1}$).

|  | Newly constructed 1887(1891)-2019 | Berkeley Earth (1873-2019) | CRUTS4.03 (1901-2018) |
|---|---|---|---|
| Maximum temperature | 0.119±0.015 | 0.099±0.010 | 0.062±0.015 |
| Minimum temperature | 0.194±0.013 | 0.156±0.010 | 0.217±0.015 |
| Mean temperature | 0.154±0.013 | 0.128±0.009 | 0.140±0.013 |

Table 5 indicates that the annual trends of newly constructed maximum (1887-2019) and minimum temperature (1891-2019) series in Tianjin are 0.119±0.015 ℃ decade$^{-1}$ and 0.194 ±0.013 ℃ decade$^{-1}$. Trend changes based on the newly constructed series are nearly consistent with those in Berkeley Earth and CRUTS4.03 on the century-long scale and these are 0.099±0.010 ℃ decade$^{-1}$ and 0.156±0.010 ℃ decade$^{-1}$, 0.062±0.015 ℃ decade$^{-1}$ and 0.217±0.015 ℃ decade$^{-1}$ respectively. The trend of the mean temperature for the newly constructed series (0.154±0.013 ℃ decade$^{-1}$) is slightly larger than those from the interpolated series from Berkeley Earth, CRUTS4.03, and Cao et al. (2013) (0.128±0.009 ℃ decade$^{-1}$, 0.140±0.013 ℃ decade$^{-1}$, and 0.098±0.017 ℃ decade$^{-1}$, respectively). The average temperature trend changes from the newly constructed series are much closer to internationally authoritative data calculations, so they are more consistent. Moreover, annual trend change in mean temperature based on newly constructed series in Tianjin is similar to that for China (Li et al., 2020c), which are 0.130±0.009 ℃ decade$^{-1}$, 0.114±0.009 ℃ decade$^{-1}$ and 0.121±0.009 ℃ decade$^{-1}$ respectively from CRUTEM4, GHCNV3 and C-LSAT (during 1900 - 2017).





## 5.2 Extreme events change trend during the last 130 years

**Table 6.** Trends in annual and seasonal temperature extremes of Cold nights (TN10p), Cold days (TX10p), Warm nights (TN90p) and Warm days (TX90p) for daily newly constructed temperatures in Tianjin (Units: d decade$^{-1}$).

|  | TN10p | TX10p | TN90p | TX90p |
|---|---|---|---|---|
| Annual | -1.454* | -0.140* | 1.196* | 0.975* |
| Spring | -1.861* | -0.508* | 1.423* | 0.959* |
| Summer | -1.483* | -0.213* | 1.443* | 1.474* |
| Autumn | -0.798* | -0.221* | 0.724* | 0.621* |
| Winter | -1.555* | 0.421* | 1.119* | 0.850* |

The asterisks indicate trend changes are significant at the 5% level. The annual time periods of TN10p and TN90p cover the 1891-2019 period, TX10p and TX90p cover 1887-2019. The time periods of TN10p and TN90p in spring and summer cover 1891-2019, those in autumn and winter 1890-2019 (winter ending in 2018), and TX10p and TX90p in four seasons all cover 1887-2019 (winter ending in 2018).

Table 6 indicates that trends of temperature extremes based on the newly constructed series are all significant at the 5% level and have much more coincident changes. The cold extremes (TN10p and TX10p) at annual and seasonal timescales express significantly decreasing trends (except for TX10p in winter), while the warm extremes (TN90p and TX90p) show increasing trends. Trends of TN10p, TX10p, TN90p and TX90p at the annual timescales are -1.454 d decade$^{-1}$, -0.140 d decade$^{-1}$, 1.196 d decade$^{-1}$ and 0.975 d decade$^{-1}$, all passed the significance test at the 5% level. For the seasonal change, trends of TN10p and TX10p in spring are the largest. They are -1.861 d decade$^{-1}$ and -0.508 d decade$^{-1}$ during the past 130 years. For TN90p and TX90p is in summer, with the amplitude of 1.443 d decade$^{-1}$ and 1.474 d decade$^{-1}$.

## 6 Data availability

The newly homogenized daily surface air temperature for Tianjin city over century-long scales are published at PANGAEA (https://doi.pangaea.de/10.1594/PANGAEA.924561, last access: 10 November



2020) under the DOI https://doi.pangaea.de/10.1594/PANGAEA.924561 (Si and Li, 2020). The dataset
contain the maximum, minimum and mean temperature time series before and after adjustment as well
as new estimates of average and extreme temperature trend change in Tianjin for the period of

1887-2019.

**7    Conclusions and discussion**
This paper documents the various procedures necessary to construct a homogenized daily maximum and
minimum temperature series since 1887 for Tianjin. These same procedures could and should be used
for other sufficiently long and complete series across the world. The newly constructed data have
reduced non-climatic errors at the daily timescale, improved the accuracy and enhanced the real climate
representation of average and extreme temperature over century-long scales.

Three sources of surface observation daily data collected by the Tianjin Meteorological Archive

have been integrated viz., Department of Industry Agency of British Concession in Tianjin covering Sep
1 1890-Dec 31 1931, Water Conservancy Commission of North China covering Jan 1 1932-Dec 31 1950,
and monthly journal sheets of Tianjin surface meteorological observation records covering Jan 1
1951-Dec 31 2019. These three have provided a good foundation for the construction of reliable
homogenized daily series by quality control of climatic range checks, climatic outlier checks and
internal consistency checks. Data extension has been undertaken in the interest of extending the length
of the series as far back as possible, but it is carried out only for the daily maximum series due to length
limitation of reference daily data.

Using the integration, quality control and extension, we detected and adjusted the statistically

significant breakpoints in the daily maximum and minimum temperature time series from an objective
perspective based on multiple reference series and statistical characteristics from homogenization
detection by means of PMT as well as sophisticated manual data processing. This temperature series
provides a set of new baseline data for the field of extreme climate change over the century-long scale
and a reference for construction of other long-term reliable daily time series in the region. The annual
trends of newly constructed maximum and minimum temperature in Tianjin are $0.119\pm0.015°C$ decade$^{-1}$
and $0.194\pm0.013°C$ decade$^{-1}$ over the last 130 years, which are similar to those from Berkeley and
CRUTS4.03. The trend of mean temperature averaged from the new series is $0.154\pm0.013°C$ decade$^{-1}$,
which is of the same order as those over the whole China (Li et al., 2020a; 2020c). The new daily data
also show improvements over the archived datasets for trend analyses of extremes. The trends in TN10p,
TX10p, TN90p and TX90p are all significant at the 5% level, and they give a much more consistent set
of trends. To some extent, changes in climate extremes can be analyzed with higher confidence using the
newly constructed daily data in this paper.

However, in the current study, there may be some systematic biases (possibly some potential

breakpoints missed) still in the adjusted time series, since metadata of Tianjin station are not consistently
available in the climatological archives over the whole century as well as not being documented during
the period before 1921. Climate data homogenization does not always follow a consistent pattern (Si et
al., 2018; 2019). It is necessary to constantly improve the existing methodology and explore new
techniques in order to obtain reliable homogenized data products. Accordingly, future work should
involve more detailed station metadata and more advanced data processing techniques to produce much
better daily datasets over century scales.
**Author contributions.** QL designed and implemented the dataset construction. PS collected the basic
and reference data sources, constructed the dataset and written the paper. QL, PS and PJ all contributed
to data analysis, discussion and writing of the paper.



**Competing interests.** The authors declare that they have no conflict of interest.
**Acknowledgements.** We thank the many people and /or institutions who contributed to the construction
of this dataset.
**Financial support.** This research has been supported by National Natural Fund projects No. 41905132
and No. 41975105.

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
