# Peer review of "Construction of homogenized daily surface air temperature for Tianjin city during 1887-2019"

_Earth System Science Data, 2020_

## Author Response (AR1)

**Reply to Anonymous Referee #1:**

Thank you very much for the valuable time devoted to this paper by the referees and the responsible editor of ESSD, as well as for our opportunity to reply to these comments. A point-by-point response to Anonymous Referee #1' comments is as follows.

**Major comments:**

1. Figure 6: I noticed that the significant breakpoint occurred in 1955 for Tmax but in 1992 for Tmin. Why did these points differ greatly? Is this due to non-climate factor such as station relocation? If so, the influence is supposed to be the same. Explanation or discussion for this interesting phenomenon will benefit the improved understanding of the newly-constructed homogenized dataset.

**Reply:Thank you for your valuable comments.**

It may be related with the physical characteristics of Tmax and Tmin themselves. The Tmin generally occurs near sunrise when calm and stable atmospheric boundary layer conditions are prevalent. Under these conditions, near surface temperature fields are strongly coupled to the local surface characteristics. On the other hand, during daylight hours (like Tmax), the boundary layer is commonly well mixed, and microclimate differences between nearby sites may be less evident. So the difference of the discontinuities between Tmax and Tmin temperature is not that uncommon (e.g. Tmin always has more discontinuities in total, see the details in Li and Dong, 2009).

As to the discontinuities, the breakpoints in Jan 1 1955 and Jan 1 1992 are both caused by station relocation, which affects the homogeneity of the temperature

elements differently. In Jan 1 1955 Tianjin meteorological station relocated from No. 22 Ziyou Road to Zunyi Road (Table 1), which is 5km north of the original site. However, the relocation of Tianjin meteorological station in 1992 was from Qixiangtai Road, Hexi district to Xidawa, Xiqing District. In addition, there are another two discontinuities in 1897 and in 1907 for Tmin (and both of these two did not affect Tmax).

**We have added this paragraph on L334—L337 in revised manuscript 'Si et al -20210311_resub.pdf'.**

Ref:

Li Q. and Dong W., 2009, Detection and Adjustment of Undocumented Discontinuities in Chinese Temperature Series Using a Composite Approach, Adv., Atmos., Sci.,26(1):143-153,doi: 10.1007/s00376-009-0143-8

2. L393-396: it seems a parodox to argue that "annual trend change in mean temperature based on newly constructed series in Tianjin is similar to that for China (Li et al., 2020c)". On the contrary, the trends derived from the other two dataset (Berkeley Earth and CRUTS4.03) are more similar the national warminig rate as shown in Table 5. The authors are suggested to clarify this point.

**Reply:Thank you for your valuable comments.**

The annual trend in Tmean from the newly constructed series in Tianjin is a little larger than that over the whole of China. We think this conforms that the result in this manuscript is reasonable. Because the trend in northern regions is more prominent than for other regions in mainland China (Li et al., 2004; Zhai et al., 2004). **Therefore,**

**according to the comments, the authors have rewritten the sentence into** 'Moreover, annual trend change in mean temperature based on the newly constructed series at Tianjin is also a little larger than that over the whole China (Li et al., 2020c), which are $0.130 \pm 0.009\,°C$ decade$^{-1}$, $0.114 \pm 0.009\,°C$ decade$^{-1}$ and $0.121 \pm 0.009\,°C$ decade$^{-1}$ respectively from CRUTEM4, GHCNV3 and C-LSAT (during 1900 - 2017).' **on L402—L406 in revised manuscript 'Si et al -20210311_resub.pdf'.**

**And the authors also added the sentence** 'It conforms to the underlying changes across China. Increasing trends in northern China are more prominent than those from other regions in mainland China (Li et al., 2004; Zhai et al., 2004).' **on L406—L407, and added two references** (Li et al., 2004; Zhai et al., 2004) **in References section on L531—L533, L622—L623 in revised manuscript 'Si et al -20210311_resub.pdf'.**

**Ref:**

Li., Q. X., Zhang., H. Z., Liu., X. N., and Huang., J. Y.: Urban heat island effect on annual mean temperature during the last 50 years in China, Theor. Appl. Climatol., 79, 165-174, https://doi.org/10.1007/s00704-004-0065-4, 2004.

Zhai., P. M., Chao., Q. C., and Zou., X. K.: Progress in China's climate change study in the 20th century, J. Geograph. Sci., 14(1): 3-11, https://doi.org/10.1007/BF02841101, 2004.

**Minor comments:**

1. L55: "representativeness" -> "better representativeness"

**Done**

2. L286: "surface observation station" -> "surface weather station"

**Done**

3. L291: "quantile matching" has been widely used in recent research associated with precipitation (Lv et al. 2020, doi:1016/j.atmosres.2019.104671), and PM2.5 (Bai et al. 2020, doi:10.5194/essd-12-3067-2020), which is suggested to be mentioned.

**Done**

4. L384: "indicates"-> "shows"

**Done**

5. L392: "internationally authoritative data calculations"? are there any references to support this argument? Further, this sentence is not logically connected with the following sentence "so they are more consistent". For instance, what does the "they" refer to? Therefore, it is suggested to be rewritten.

**According to the Major Comments 2, we have deleted this sentence '**The average temperature trend changes from the newly constructed series are much closer to internationally authoritative data calculations, so they are more consistent.**' on L400—L402, and rewritten the second half of the paragraph into** 'Moreover, annual trend change in mean temperature based on the newly constructed series at Tianjin is also a little larger than that over the whole China (Li et al., 2020c), which are $0.130\pm0.009$ ℃ decade$^{-1}$, $0.114\pm0.009$ ℃ decade$^{-1}$ and $0.121\pm0.009$ ℃ decade$^{-1}$

respectively from CRUTEM4, GHCNV3 and C-LSAT (during 1900 - 2017). It conforms to the underlying changes across China. Increasing trends in northern China are more prominent than those from other regions in mainland China (Li et al., 2004; Zhai et al., 2004).' **on L402—L407 in revised manuscript 'Si et al -20210311_resub.pdf'.**

6. L404: "indiates"-> "shows" or "illustrates"

**Done**

7. L409: grammar errors in "all passed"

**Done**

8. L410: "trends of TN10p and TX10p in spring are the largest. They are" -> "the negative trends of TN10p and TX10p in spring are the largest, reaching up to be"

**Done**

9. L422: it seems a little strong tone to argue "These same procedures could and should be use", which can be softened, since there are large room to improve the procedures for data homogeity. More importantly, it remains unknown whether the procedure developed here can be genalized or applied to other regions, which merits further investigation.

**The word 'procedure' refers to the steps of constructing a long and complete climate time series, rather than the techniques and methods. To avoid the**

confusion, we have modified this sentence into 'These **similar** procedures could

and should be used for other sufficiently long and complete series across the world.'

on L433 in revised manuscript 'Si et al -20210311_resub.pdf'.

**Reply to Anonymous Referee #2:**

Thank you very much for the valuable time devoted to this paper by the referees and responsible editor of ESSD, as well as for our opportunity to reply to these comments. A point-by-point response to Anonymous Referee #2' comments is as follows.

**Specific comments:**

1. Please provide the information about SAT stations (location, surroundings, rural, towns or city, as well as altitude).

**Reply:Thank you for your suggestions.**

We have provided the information about SAT stations in the **new Table 3 (L287-L288)**, and added some description sentence on **L279** in revised manuscript '**Si et al -20210311_resub.pdf**'.

2. Provide a correlation analysis and other measure to convince the use these data.

**Reply:Thank you for your suggestions.**

It is difficult to find much observational data surrounding Tianjin before the 1950s, so in the establishing the reference series, we employed the station series or the interpolated temperature series using neighboring grid boxes from three global land surface temperature observation series (CRUTS4.03, GHCNV3 and Berkeley Earth). For monthly reference series, we established two types using a weighted average method. One was based on the combination of the interpolated temperature series from Berkeley Earth and CRUTS4.03 and station series from GHCNV3 data for

Tianjin site and the other was based on the interpolated temperature series (they are the selected 9 stations) from Berkeley Earth data only. It is worth mentioning that the reference series established by the 3 datasets are very consistent, which reflects the consistency of air temperature variation in this region among different datasets. All the interpolated station series used in this current study have much higher correlations with each other, with the correlation coefficients all higher than 0.95.

3. Justify why 300km is reasonable.

**Reply:Thank you for your comments.**

In the process of data homogenization, the standard practice is to select 3-5 surrounding stations as the reference stations (the correlation coefficient between these stations and candidate stations is greater than 0.8, and the distance between them is 300-500km) for discontinuous point detection and adjustment, establish a relatively homogenous reference series that can represent the local climate change characteristics for the candidate stations, and detect the homogeneity of the difference series between the candidate and the reference series. However, this ideal depends on the density of the network. In the current study, we found 9 stations (including 54527 itself) within 300 km as the basis of reference series construction, which shows that the reference series establishment in this area is relatively reasonable, and using as many as possible stations can offset some inhomogeneity of a station that might affect the reasonableness of the reference series as far as possible.

4. Please explore why use monthly and daily reference series respectively for the extreme temperature series? The extreme temperature series is monthly or daily series?

**Reply:Thank you for your comments.**

Because daily temperatures vary on relatively small spatial scales and are influenced by local processes that are complex and nonlinear, homogenization detection for daily data is difficult. The **monthly reference series** were used to **detect breakpoints** for the extreme temperature series in the process of PMT, and the **daily reference series** was used to **adjust breakpoints** in the process of QM-adjustment. **The extreme temperature series are daily series.** What we have done in this current study is the same as in Xu et al (2013).

Ref:

Xu, W. Q., Li, Q. X., Wang, X. L., Yang, S., Cao, L. J., and Feng, Y.: Homogenization of Chinese daily surface air temperatures and analysis of trends in the extreme temperature indices, J. Geophys. Res. Atmos., 118, http:// doi.org/10.1002/jgrd.50791, 2013.

**Minor comments:**

1. L248 For Table 2, why there are four data sources shown here, the Table 1 is puzzle?

There are **three** data sources in Table 2, which are CRUTS4.03, Berkeley Earth and GHCNV3, respectively.

The CRUTS4.03 and GHCNV3 data meet the requirements of this current study and both are on monthly timescales, but the Berkeley Earth data are on monthly and daily

timescales. Table 1 only introduced the metadata information of Tianjin meteorological observation station.

We **made some revisions** (changed all "Berkeley Earth" into "BE") to show this clearer in revised manuscript '**Si et al -20210311_resub.pdf**'.

2. L284 "Berkeley-daily data" should be "Berkeley Earth-monthly data".

It should be "**BE-daily data**". We have changed on **L286** in revised manuscript '**Si et al -20210311_resub.pdf**'**.**